# Antibiotic Use during Pregnancy in South Korea Using 2011–2020 National Health Insurance Claims Data

**DOI:** 10.3390/antibiotics12081242

**Published:** 2023-07-28

**Authors:** Jungmi Chae, Jun Yong Choi, Bongyoung Kim, Dong-Sook Kim

**Affiliations:** 1Department of Research, Health Insurance Review and Assessment Service, Wonju 26465, Republic of Korea; choh1014@hira.or.kr; 2Department of Internal Medicine, Yonsei University College of Medicine, Seoul 03722, Republic of Korea; seran@yuhs.ac; 3Department of Internal Medicine, Hanyang University College of Medicine, Seoul 04763, Republic of Korea; 4Department of Health Administration, College of Health, Kongju National University, Gongju 32588, Republic of Korea

**Keywords:** pregnancy, antibiotics, outpatients

## Abstract

Background: Since antimicrobial overuse and misuse can have substantial impacts on both public health and fetal well-being, it is essential to gain comprehensive insights into antimicrobial consumption patterns in pregnant women. This study aims to demonstrate antimicrobial utilization in pregnant women. Methods: We conducted a population-based cohort study using National Health Insurance claims data from January 2009 to December 2020 in South Korea. The target population was pregnancies in women aged 15–45 years who gave birth between 2011 and 2019. The outcome measure was the percentage of antibiotic prescriptions by trimester, subgroup, diagnostic category, and therapeutic category. Antibiotics were defined as J01 in the WHO ATC/DDD classification. To analyze the factors that influenced antibiotic prescriptions, we conducted multivariate logistic regression analysis. Results: Antibiotics were prescribed at least once in 1,808,588 (50%) of the 3,614,478 pregnancies. The proportion of deliveries with exposure to antibiotic therapy during pregnancy increased from 48% in 2011 to 54.8% in 2020. The prescription rate of antibiotics was highest in mothers younger than 25 years old, and it was lowest in participants aged 30–34 years. Also, antibiotic use was highest in the first trimester (30%) and lowest in the second trimester (18.2%). The most commonly used class was J01D (other beta-lactam antibacterials), which includes cephalosporins, and it accounted for 57% of the prescribed antibiotics. An increased probability of being prescribed antibiotics was associated with those younger than 25 years old, insurance (Medical Aid), hospitalization experience, an increase in physician visits, and comorbidities. For comorbidities, the OR was higher for respiratory tract infection (RTI), sexually transmitted infection, and urinary tract infection compared to those without disease, whereas it was decreased for diabetes and epilepsy. Conclusion: The prescribing rate of antibiotics to pregnant women in South Korea has remained stably high. Also, an increase in the use of broad-spectrum beta-lactam penicillin was noted. It is necessary to monitor antibiotics in pregnant women generally in accordance with recommendations.

## 1. Introduction

Antimicrobial resistance (AMR) is rapidly becoming one of the most significant global health challenges, posing severe threats to healthcare systems and endangering patient outcomes worldwide [1]. The overuse and misuse of antibiotics have been identified as primary drivers behind the alarming rise in resistant pathogens, leading to heightened mortality rates, prolonged hospital stays, increased medical costs, and a decline in the effectiveness of essential antimicrobial therapies [2]. To address this critical issue, curbing the inappropriate use of antibiotics has become an urgent public health priority. Prominent global health organizations, such as the World Health Organization (WHO), have stressed the urgency of developing an international plan of action that underscores the responsible use of antibiotics [3].

To combat antimicrobial resistance effectively, it is essential to gain comprehensive insights into antimicrobial consumption patterns across diverse populations [4,5]. While the broader population’s antibiotic usage is undoubtedly crucial, one specific demographic, pregnant women, warrants special attention. Pregnancy is a period marked by heightened susceptibility to infections due to physiological changes and immune system adaptations [6]. Consequently, pregnant women frequently encounter a myriad of infections, and antimicrobial prescriptions are often warranted to safeguard both maternal and fetal health [7,8,9,10,11,12,13,14,15].

However, the indiscriminate use of antibiotics during pregnancy could exacerbate resistance rates, with far-reaching consequences for not only the current generation but also for future generations. Also, evidence indicates that antimicrobial agents can cross the placental barrier, exposing the developing fetus to the adverse effects of antibiotics, resulting in several negative health conditions [16,17,18,19,20,21,22,23]. 

Since antimicrobial overuse and misuse can have substantial impacts on both public health and fetal well-being, numerous studies have investigated antibiotic use during pregnancy in Europe, the United States, and China [12,13,14,15,24,25,26,27]. These studies have revealed that antibiotics are prescribed to approximately 20–50% of pregnant women [24,25,26,27]. However, despite the valuable data available from international research, there is a lack of studies that specifically analyze trends in antibiotic utilization among pregnant women in Korea.

This study aims to demonstrate antimicrobial utilization in pregnant women. By analyzing prescription patterns, problems can be identified, and opportunities associated with antimicrobial stewardship for pregnant women could be developed.

## 2. Results

### 2.1. Pregnancy and Antibiotic Use by Year and Age

During the 10-year study period from 2011 to 2020, we identified 3,614,478 deliveries. The mean age at delivery was 32.2 (±4.12) years. In total, the number of pregnant women decreased from 420,001 in 2011 to 259,610 in 2020. Although the number of antibiotic prescriptions during pregnancy decreased from 201,514 in 2011 to 142,206 in 2020, the prescription rate of antibiotics increased from 48% in 2011 to 54.8% in 2020 (Figure 1A).

By age, the prescription rate of antibiotics was highest in women aged 24 or younger (55.6%), followed by women aged 40 or more (52.1%), and it was lowest in women aged from 30 to 34 (47.4%). The antibiotic prescription rate was highest in those aged 24 or younger, and this rate increased from 51.9% in 2012 to 61.6% in 2019. In association with an increased number of deliveries in these age groups, the number of pregnancies with antibiotic prescriptions increased from 35,568 (48.3%) in 2011 to 50,619 (55.3%) in 2020 for women aged 35–39 and increased from 5102 (50.7%) in 2011 to 9358 (56.6%) in 2020 for women aged 40 or more (Figure 1B).

### 2.2. Antibiotic Use by Trimester

Table 1 shows antibiotic use by trimester. The prescription rate of antibiotics was highest (30%) in the first trimester and lowest (18.2%) in the second trimester, with a slightly higher rate (18.6%) in the third trimester. By age, in the first trimester, the prescription rate of antibiotics was highest in women aged 24 or younger (34.2%) and lowest in women aged from 30 to 34 (29%).

The prescription rate of antibiotics in women with respiratory tract infections was 68.6% in the total period, 44.5% in the first trimester, 25.4% in the second trimester, and 25.7% in the third trimester, while for women without respiratory tract infections, the antibiotic prescription rate was 32% in the total period, 15.9% in the first trimester, 11.2% in the second trimester, and 11.6% in the third trimester.

The antibiotic prescription rate in the first trimester steadily increased from 28.3% in 2012 to 34.7% in 2020, while antibiotic use in the second trimester rose until 2019 and dropped in 2020.

### 2.3. Use of Antibiotics according to ATC Level

Table 2 shows antibiotic use by drug class in the total period and each trimester. The most commonly used ATC third level was J01D (other beta-lactam antibacterials), which includes cephalosporins, and it accounted for 57% of the prescribed antibiotics, followed by J01C (beta-lactam antibacterials, penicillins) with 45.2%, and J01F (macrolides, lincosamides, and streptogramins) with 20.7%. In particular, in the first trimester, J01M (quinolone antibacterials) accounted for 9.7% of prescriptions and J01A (tetracyclines) for 6.2%, whereas their use was lower than 1% in the second and third trimesters.

By ATC level 4, the prescription rate of J01CR (combination of penicillins, including beta-lactamase inhibitors) was 33.4%, the prescription rate of antibiotics belonging to J01FA (macrolides, including erythromycin, roxithromycin, and azithromycin) was 19.6%, and the prescription rate of third-generation cephalosporin was 12.8%.

The proportions of prescriptions in categories J01A, J01F, and J01X consistently increased throughout the study period, while those of J01C and J01M decreased gradually (Table 2).

### 2.4. Factors That Influenced Antibiotic Prescriptions

Table 3 summarizes the results of the logistic regression analysis performed to explore variables associated with the likelihood of antibiotic use during pregnancy. Those younger than 25 years; those who received Medical Aid (compared to those covered by National Health Insurance); and those with hospitalization experience, a higher number of physician visits, and comorbidities had a higher probability of being prescribed antibiotics, while emergency department visits were associated with lower odds.

When compared to those aged 40 years or more, the odds ratio (OR) of antibiotic use was 1.15 (95% confidence interval [CI]: 1.13–1.17) in those younger than 25 years, and the OR of being prescribed antibiotics was higher than 1 in the other age groups. The OR was 3.06 (95% CI: 3.03–3.08) in patients with hospitalization experience, whereas it was 0.98 (95% CI: 0.97–0.99) in those who visited the emergency department.

Regarding comorbidities, high ORs were found in patients with respiratory tract infections (OR 3.97, 95% CI: 3.95–3.99), sexually transmitted infections (OR 4.13, 95% CI: 4.07–4.19), and urinary tract infections (OR 3.05, 95% CI: 2.96–3.10) compared to those without diseases, whereas the likelihood of antibiotic use was lower in those with diabetes (OR 0.82, 95% CI: 0.81–0.84) and epilepsy (OR 0.82, 95% CI: 0.77–0.87).

## 3. Methods

### 3.1. Data Source 

We conducted a retrospective population-based study using the Korean National Health Insurance claims database from March 2010 through December 2020. South Korea has a single-payer health insurance system that provides coverage for all citizens and reimburses providers on a fee-for-service basis. As of 2020, this database contained information on both in-hospital and outpatient visits from a population of 51.8 million. The database includes demographic characteristics, medical conditions (diagnoses), healthcare utilization (visit dates, tests, procedures, length, and spending), and medicine use (product name, ingredient name, dose, days of therapy, and spending). All claims have been submitted electronically since 2007, and all data files (e.g., type of medical facilities and patients’ demographic files) could be linked by unique patient identification numbers.

Medical conditions are classified according to the International Classification of Diseases, Tenth Revision (ICD-10). The overall positive predictive value of diagnosis records was found to be 82% in a validation study comparing our database with electronic medical records. Medicines are recorded using international nonproprietary names and the codes of individual products. In South Korea, most people tend to visit medical institutions to obtain prescriptions rather than purchasing over-the-counter medications at pharmacies, so most medicine use in the total population is recorded through electronic claims in the health insurance database. 

### 3.2. Definition of the Pregnancy Period and Exposure to Antibiotics 

Our cohort included all pregnancies in women resulting in live births in South Korea from 1 January 2011 to 31 December 2020, and the pregnancies were identified by procedure codes for delivery. The study population included all women aged 15 to 45 years at delivery, and all inpatient and outpatient medical encounters in this population from March 2010 through December 2020 were used in the analysis. The medical institutions included in the data were tertiary hospitals, secondary hospitals, general hospitals, nursing homes, clinics, and public health centers. 

The time points of natural childbirth or cesarean section were regarded as delivery. We estimated the pregnancy period using the following formula: (birth date—gestational period [40 weeks]) + 2 weeks. We defined the following four pregnancy-related periods: the prepartum period (up to three months before pregnancy) and three pregnancy periods, including the first trimester (0–90 days), second trimester (91–181 days), and third trimester (182 days to delivery). We excluded pregnancies ending in the second trimester. Pregnancy was defined as a date prior to 294 days from the date of delivery or surgery.

The study population was selected as shown in Figure 2. Eventually, the final number of women at delivery whose pregnancy was maintained until the third trimester was 3,614,478.

We defined antibiotics as J01 (systemic use of antimicrobials) according to the WHO Anatomical Therapeutic Chemical (ATC) drug classification system [28]. We used drug classes of both the third level of the ATC system (pharmacological subgroups) and the fifth level of the ATC system (chemical substances). We analyzed the exposure to antibiotics during pregnancy by the ATC-3 level and ATC-4 level.

### 3.3. Outcome Measures and Covariates

The outcome measure was prescriptions of antibiotics. The explanatory variables were classified into the following categories: maternal demographic status (age (≤24, 25–29, 30–34, and 35–39), type of insurance (Health Insurance, Medical Aid), presence of multiple births (yes, no), maternal conditions (hypertension, diabetes, renal diseases, gastrointestinal diseases, and depression), infectious diseases (respiratory tract infections, sexually transmitted infections, and urinary tract infections), and general markers of the burden of illness (emergency department visits, hospital admissions, and number of outpatient visits). Medical conditions were classified using primary or secondary diagnoses according to the ICD-10 (Appendix A).

### 3.4. Statistical Analysis

We present descriptive statistics on the prevalence of antibiotic use during the prenatal period and pregnancy. We used the chi-square test to test for time trends.

We also used multivariable logistic regression to generate crude and adjusted odds ratios (ORs) and 95% confidence intervals (CIs) for maternal characteristics associated with antibiotic prescriptions during pregnancy. We conducted a sensitivity analysis according to trimester. All analyses were performed using SAS Enterprise 9.4 (SAS Institute, Cary, NC, USA). This study was approved by the Institutional Review Board of Health Insurance Review & Assessment Service (2021-016; 5 March 2021).

## 4. Discussion

To our knowledge, this is the first population-level study conducted in South Korea to investigate the status of antibiotic use in pregnant women in Korea. It included 3,614,478 deliveries from 2011 to 2020, and 50% of pregnancies had at least one antibiotic prescription. The prescription rate of antibiotics increased from 48% in 2011 to 54.8% in 2020.

The results of this study are similar to those of studies that investigated the prescription rates of antibiotics in pregnant women in other developed countries. According to a study by Petersen conducted in 2010 of 114,999 women in the UK from 1992 to 2007, the probability of being prescribed antibiotics during pregnancy was found to be 14% [26]. Also, in a study in the Netherlands, 20.8% of women were prescribed at least one antibiotic during pregnancy [13]. According to existing studies, region, age, and social status were found to be associated with antibiotic prescriptions [24]. In this study, maternal age younger than 25 years at birth, experience of hospitalization, sexually transmitted infections, respiratory system infections, and urinary tract infections, compared to those without the corresponding experiences or infections, were identified as factors strongly associated with a higher likelihood of antibiotic prescriptions during pregnancy. This result is consistent with a previous study reporting that previous childbirth and maternal asthma were associated with the use of antibiotics for respiratory tract infections during pregnancy [22].

The most commonly used ATC third level category was J01D (other beta-lactam antibacterials), and in the first trimester, the use of J01M (quinolone antibacterials) and J01A (tetracyclines) was higher than in the first and second trimesters. Since only 10% of medications have sufficient data related to safe and effective use in pregnancy, it is crucial to analyze the real-world use of antibiotics in pregnant women. Thus, future research might be needed to investigate the evidence of the risks and benefits of using antibiotics for which safety and efficacy information is not available from randomized controlled trials.

The main strength of the present study is that it is based on a high-quality data source, the Korean National Health Insurance claims database, which is the most representative health data in South Korea. South Korea has a health insurance system that provides coverage for all citizens and reimburses providers on a fee-for-service basis; the number of doctor consultations per person was high. As prescriptions for mild diseases are also given at medical institutions in Korea, the records for these institutions can be considered to include all antibiotics except for those used for medical services that are not covered by Korean National Health Insurance, such as cosmetic procedures. Moreover, since the National Health Insurance claims database includes both fee-for-service deliveries and deliveries claimed through the DRG system, it can be considered to include all data regarding women who gave birth.

Nevertheless, this study has the following limitations. First, it only included women who delivered babies, and those who had abortions or stillbirths were excluded. Thus, since cases in which babies died due to antibiotic prescriptions during the early stages of pregnancy were not included, there is a possibility of under-estimation. Second, although culture test status could be checked as part of the National Health Insurance claims data, as the results of laboratory tests were submitted for review, they were not analyzed. It will be necessary for future studies to include an analysis of culture test results to determine whether appropriate antibiotics were chosen. Third, although antibiotic administration during pregnancy and birth may be influenced by social and lifestyle-related factors, our analysis used an administrative dataset; thus, information was lacking on social factors. Since understanding these risk factors may guide preventive strategies in order to avoid the unnecessary use of antibiotics, further research would need to investigate both social and clinical factors.

## 5. Conclusions

In conclusion, the prescription rate of antibiotics for pregnant women in South Korea has remained stably high. An increase in the use of broad-spectrum beta-lactam penicillin was also noted. Therefore, it is necessary to monitor antibiotic use in pregnant women in accordance with recommendations.

## Figures and Tables

**Figure 1 antibiotics-12-01242-f001:**
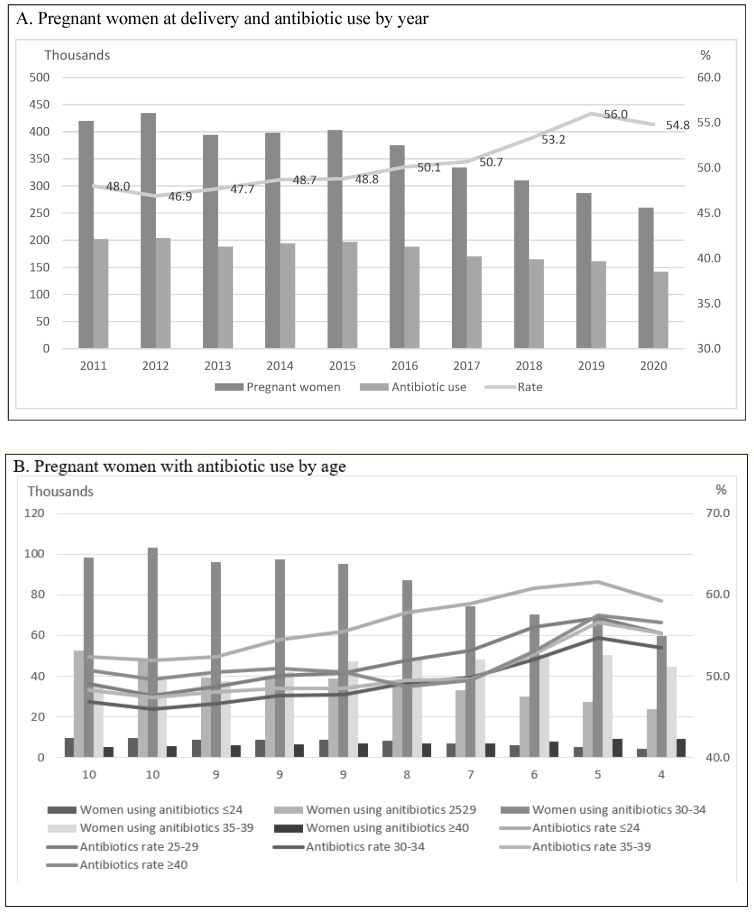
Trends in the number of women at delivery and antibiotic use (thousands).

**Figure 2 antibiotics-12-01242-f002:**
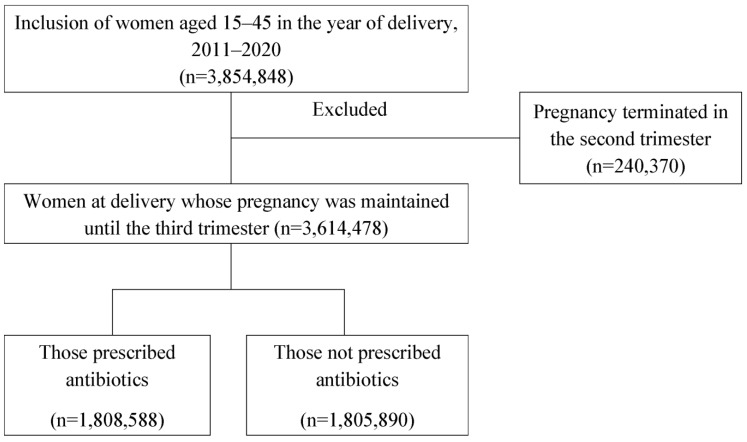
Selection of the study population.

**Table 1 antibiotics-12-01242-t001:** Pregnant women and antibiotic use by trimester (%).

	Total Pregnancies	No. of Pregnancies with Antibiotic Use							
		Total Period		First Trimester		Second Trimester		Third Trimester	
Number of women at delivery	3,614,478	1,808,588	(50.0)	1,083,040	(30.0)	657,927	(18.2)	670,999	(18.6)
Age									
≤24	138,854	77,224	(55.6)	47,519	(34.2)	29,002	(20.9)	30,585	(22.0)
25–29	719,653	367,777	(51.1)	221,857	(30.8)	131,404	(18.3)	140,876	(19.6)
30–34	1,742,929	849,997	(48.8)	505,538	(29.0)	300,033	(17.2)	319,713	(18.3)
35–39	877,886	443,228	(50.5)	266,349	(30.3)	167,467	(19.1)	157,499	(17.9)
≥40	135,156	70,362	(52.1)	41,777	(30.9)	30,021	(22.2)	22,326	(16.5)
Respiratory tract infections (RTIs)									
Antibiotic use for RTIs	1,778,437	1,220,196	(68.6)	791,970	(44.5)	452,408	(25.4)	457,400	(25.7)
Antibiotic use for non-RTIs	1,836,041	588,392	(32.0)	291,070	(15.9)	205,519	(11.2)	213,599	(11.6)
Year									
2011	420,001	201,514	(48.0)	121,453	(28.9)	69,452	(16.5)	73,960	(17.6)
2012	434,089	203,654	(46.9)	122,965	(28.3)	72,979	(16.8)	70,025	(16.1)
2013	394,219	187,863	(47.7)	112,086	(28.4)	68,360	(17.3)	66,033	(16.8)
2014	397,564	193,551	(48.7)	113,998	(28.7)	71,765	(18.1)	70,013	(17.6)
2015	402,933	196,782	(48.8)	116,368	(28.9)	72,891	(18.1)	71,521	(17.8)
2016	374,981	187,730	(50.1)	109,220	(29.1)	69,779	(18.6)	72,128	(19.2)
2017	334,336	169,641	(50.7)	99,616	(29.8)	61,859	(18.5)	67,015	(20.0)
2018	309,528	164,756	(53.2)	98,614	(31.9)	60,337	(19.5)	65,072	(21.0)
2019	287,217	160,891	(56.0)	98,595	(34.3)	59,825	(20.8)	63,481	(22.1)
2020	259,610	142,206	(54.8)	90,126	(34.7)	50,688	(19.5)	51,756	(19.9)

RTI, respiratory tract infection.

**Table 2 antibiotics-12-01242-t002:** Antibiotics prescribed in pregnancy per trimester by ATC level (thousands, %).

	Delivery		By Trimester						Antibiotic Use for RTI	
			First Trimester		Second Trimester		Third Trimester			
Antibiotic use	1809	(100)	1083	(100)	658	(100)	671	(100)	1220	(100.0)
ATC-3 level										
J01A (Tetracyclines)	69	(3.8)	68	(6.2)	1	(0.2)	1	(0.1)	33	(2.7)
J01B (Amphenicols)	0.2	(0.0)	0.2	(0.0)	0.02	(0.0)	0.01	(0.0)	0.16	(0.0)
J01C (Beta-lactam antibacterials, penicillins)	818	(45.2)	411	(38.0)	265	(40.2)	278	(41.4)	643	(52.7)
J01D (Other beta-lactam antibacterials)	1031	(57.0)	511	(47.2)	346	(52.6)	371	(55.3)	711	(58.3)
J01E (Sulfonamides and trimethoprim)	9	(0.5)	7	(0.7)	1	(0.1)	1	(0.1)	6	(0.5)
J01F (Macrolides, lincosamides, and streptogramins)	374	(20.7)	221	(20.4)	106	(16.1)	79	(11.8)	263	(21.5)
J01G (Aminoglycoside antibacterials)	136	(7.5)	124	(11.5)	6	(0.9)	7	(1.1)	82	(6.7)
J01M (Quinolone antibacterials)	109	(6.1)	105	(9.7)	2	(0.4)	2	(0.3)	77	(6.3)
J01R (Combinations of antibacterials)	11	(0.6)	10	(0.9)	1	(0.1)	0.5	(0.1)	6.3	(0.5)
J01X (Other antibacterials)	166	(9.2)	84	(7.7)	40	(6.0)	52	(7.8)	87	(7.1)
ATC-4 level										
J01CA (Penicillins with extended spectrum)	348	(19.3)	151	(13.9)	137	(20.8)	141	(21.0)	248	(20.3)
J01CR (Combinations of penicillins, including beta-lactamase inhibitors)	542	(30.0)	311	(28.7)	164	(25.0)	174	(26.0)	461	(37.8)
J01DB (First-generation cephalosporins)	357	(19.7)	160	(14.8)	141	(21.5)	155	(23.0)	223	(18.3)
J01DC (Second-generation cephalosporins)	678	(37.5)	377	(34.8)	226	(34.4)	236	(35.2)	492	(40.4)
J01DD (Third-generation cephalosporins)	176	(9.8)	92	(8.5)	68	(10.4)	72	(10.8)	130	(10.7)
J01FA (Macrolides)	344	(19.0)	197	(18.2)	103	(15.6)	75	(11.3)	239	(19.5)
J01GB (Other aminoglycosides)	136	(7.5)	124	(11.5)	6	(0.9)	7	(1.1)	82	(6.7)
J01MA (Fluoroquinolones)	109	(6.1)	105	(9.7)	2	(0.4)	2	(0.3)	77	(6.3)
J01XD (Imidazole derivatives)	135	(7.5)	78	(7.2)	33	(5.0)	32	(4.8)	72	(5.9)

RTI, respiratory tract infection.

**Table 3 antibiotics-12-01242-t003:** Factors that influenced antibiotic use during pregnancy (thousands, %).

	Pregnancy (3,614,478)	Antibiotic Use (1,808,588)	OR (95% CI)
Age			
≤24	139	77 (4.3)	1.15 (1.13–1.17)
25–29	720	368 (20.3)	0.95 (0.94–0.96)
30–34	1743	850 (47.0)	0.88 (0.87–0.89)
35–39	878	443 (24.5)	0.92 (0.90–0.93)
≥40	135	70 (3.9)	-
Health insurance type			
National Health Insurance	3597	1798 (99.4)	-
Medical Aid	18	11 (0.6)	1.25 (1.21–1.29)
Multiple birth			
No	3558	1772 (98.0)	-
Yes	57	37 (2.0)	1.45 (1.42–1.48)
Experience of hospitalization			
No	3131	1454 (80.4)	-
Yes	483	355 (19.6)	3.06 (3.03–3.08)
Emergency department visit			
No	3400	1669 (92.3)	-
Yes	214	139 (7.7)	0.98 (0.97–0.99)
Physician visit, days (mean)	34.97	18.75	1.05 (1.05–1.05)
Comorbidity in pregnancy period			
Respiratory tract infection	1778	1220 (67.5)	3.97 (3.95–3.99)
Asthma	740	534 (29.5)	1.26 (1.26–1.27)
Gastrointestinal disease	774	511 (28.3)	1.65 (1.64–1.66)
Hypertension	228	124 (6.9)	1.00 (0.99–1.01)
Sexually transmitted infection	131	103 (5.7)	4.13 (4.07–4.19)
Migraine or headache	135	88 (4.9)	1.19 (1.17–1.20)
Urinary tract infection	52	40 (2.2)	3.03 (2.96–3.10)
Diabetes	66	37 (2.0)	0.82 (0.81–0.84)
Atopic dermatitis	53	31 (1.7)	0.99 (0.97–1.01)
Renal disease	9	5 (0.3)	1.13 (1.08–1.19)
Epilepsy	5	3 (0.2)	0.82 (0.77–0.87)
Drug dependent disease	1	1 (0)	1.26 (1.10–1.45)
Allergy	113	69 (3.8)	1.05 (1.04–1.06)
Year			
2011	420	202 (11.2)	-
2012	434	204 (11.3)	0.90 (0.89–0.91)
2013	394	188 (10.4)	0.92 (0.91–0.93)
2014	398	194 (10.7)	0.93 (0.92–0.94)
2015	403	197 (10.9)	0.94 (0.93–0.95)
2016	375	188 (10.4)	0.94 (0.93–0.95)
2017	334	170 (9.4)	0.93 (0.92–0.94)
2018	310	165 (9.1)	0.97 (0.96–0.98)
2019	287	161 (8.9)	1.04 (1.03–1.06)
2020	260	142 (7.8)	1.08 (1.07–1.09)

## Data Availability

https://opendata.hira.or.kr/or/orb/bigInfo.do.

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
