# Peer review of "Antibiotic Use during Pregnancy in South Korea Using 2011–2020 National Health Insurance Claims Data"

_antibiotics, 2023, doi:10.3390/antibiotics12081242_

Round 1

Reviewer 1 Report

I have reviewed the article titled: “Antibiotic Use during Pregnancy in South Korea using 2011-2 2020 National Health Insurance Claims Data” It is an article with an interesting topic related to the increase of consumption of antibiotics in pregnant women in South Korea during 10 years. However, there are some observations listed following:

1.   The abstract background is insufficient; the authors should add more background.

2.   I consider that the introduction is poor. They could provide information about the antibiotics tested and what their effects are on the health of pregnant women.

3.     There are words throughout the text that have a hyphen that should not be used, for example, pop-ulation (line 80), en-counters (line 81), medi-cal (line 82), sta-tus (line 182), etc.

4.     Format the tables, the information is piled up and it is difficult to understand them. Improve the resolution of table 3.

5.     What is the real impact of this work? Just to mention that there is an increase in the consumption of antibiotics in pregnant women?   

Bookstaver, P. B., Bland, C. M., Griffin, B., Stover, K. R., Eiland, L. S., & McLaughlin, M. (2015). A Review of Antibiotic Use in Pregnancy. Pharmacotherapy, 35(11), 1052–1062. https://doi.org/10.1002/phar.1649, Gamberini, C.; Donders, S.; Al-Nasiry, S.; Kamenshchikova, A.; Ambrosino, E. Antibiotic Use in Pregnancy: A Global Survey on Antibiotic Prescription Practices in Antenatal Care. Antibiotics 2023, 12, 831. https://doi.org/10.3390/antibiotics12050831

Minor editing of English language required

Author Response

Response to the reviewer #1’s comment

I have reviewed the article titled: “Antibiotic Use during Pregnancy in South Korea using 2011-2 2020 National Health Insurance Claims Data” It is an article with an interesting topic related to the increase of consumption of antibiotics in pregnant women in South Korea during 10 years. However, there are some observations listed following:

  1. The abstract background is insufficient; the authors should add more background.

Author’s Response: We revised the background as follows.

Due to antimicrobial overuse and misuse can substantial impact on both public health and fetal well-being, it is essential to gain comprehensive insights into antimicrobial consumption pat-terns in pregnant women. This study aims to demonstrate antimicrobial utilization in pregnant women.

  1. I consider that the introduction is poor. They could provide information about the antibiotics tested and what their effects are on the health of pregnant women.

 Author’s Response: According to the reviewer’s comment, we radically revised the introduction section. We emphasized the adverse effects of misuse and overuse of antibiotics in pregnant women in this round.

  1. There are words throughout the text that have a hyphen that should not be used, for example, pop-ulation (line 80), en-counters (line 81), medi-cal (line 82), sta-tus (line 182), etc.

 Author’s Response: Thank you for the time and effort you invested in thoroughly reviewing this manuscript. According to your comment, we revised.

  1. Format the tables, the information is piled up and it is difficult to understand them. Improve the resolution of table 3.

 Author’s Response: In order to minimize the possibility of misunderstanding, we removed the forest plot and rearranged the variables.

  1. What is the real impact of this work? Just to mention that there is an increase in the consumption of antibiotics in pregnant women?   

Author’s Response: We understand your concern. We think the value of our study is providing national antibiotic consumption data.

As well known, to combat antimicrobial resistance effectively, it is essential to gain comprehensive insights into antimicrobial consumption patterns across diverse populations. Due to antimicrobial overuse and misuse can substantial impact on both public health and fetal well-being, numerous studies have investigated antibiotic use during pregnancy in several countries. However, despite the valuable data available from international research, there is a lack of studies that specifically analyze the trends in antibiotic utilization among pregnant women in Korea.

This is the main reason why we conducted the study. We aimed to demonstrate antimicrobial utilization in pregnant women.

Reviewer 2 Report

This paper on the antibiotic use during pregnancy in South Korea is interesting, well designed and the results are well reported.

I just have a few little things that are not clear to me:  are you sure that the percentage (50.7 and 56.6) reported at line 131 are perfectly equal to the percentage reported at line 132?

At line 188 you cited a paper of Petersen and at the end of paragraph you reported the number 13 without brackets. The paper of Petersen is reported in References with the number 4, I don't understand what is 13.

Moreover at line 191 there is number 2 without brackets and its meaning is not understood.

Author Response

  1. This paper on the antibiotic use during pregnancy in South Korea is interesting, well designed and the results are well reported. I just have a few little things that are not clear to me:  are you sure that the percentage (50.7 and 56.6) reported at line 131 are perfectly equal to the percentage reported at line 132?

Author’s Response: Thank you for your comments. We revised as following:

In association with an increased number of deliveries in these age groups, the number of pregnancies with antibiotic prescriptions increased from 35,568 (48.3%) in 2011 to 50,619 (55.3%) in 2020 for women aged 35-39 and increased from 5,102 (50.7%) in 2011 to 9,358 (56.6%) in 2020 for women aged 40 or more (Figure 2B).

  1. At line 188 you cited a paper of Petersen and at the end of paragraph you reported the number 13 without brackets. The paper of Petersen is reported in References with the number 4, I don't understand what is 13.

Author’s Response: We would like to thank the reviewer for improving the manuscript. It looks like the sentence was deleted by mistake. We added as following:

Also, in the study of the Netherlands, 20.8% of the women were prescribed at least one antibiotic during pregnancy.

  1. Moreover at line 191 there is number 2 without brackets and its meaning is not understood.

Author’s Response: Sorry for our mistake. We fixed the typo in this round.

Reviewer 3 Report

In my opinion, this epidemiological study does not add to the body of literature in the field of antibiotic consumption. I do not believe that published results would be of interest to other scientists in this field worldwide. Furthermore, English language needs to be improved, and so could references. 

Needs improvement 

Author Response

In my opinion, this epidemiological study does not add to the body of literature in the field of antibiotic consumption. I do not believe that published results would be of interest to other scientists in this field worldwide. Furthermore, English language needs to be improved, and so could references.

Author’s Response: We understand your concern. However, our study can play an important role in the aspect that provides national antibiotic consumption data.

As well known, to combat antimicrobial resistance effectively, it is essential to gain comprehensive insights into antimicrobial consumption patterns across diverse populations. Due to antimicrobial overuse and misuse can substantial impact on both public health and fetal well-being, numerous studies have investigated antibiotic use during pregnancy in several countries. However, despite the valuable data available from international research, there is a lack of studies that specifically analyze the trends in antibiotic utilization among pregnant women in Korea.

This is the main reason why we conducted the study. We aimed to demonstrate antimicrobial utilization in pregnant women.

We already got an English editing service, however, if necessary, we will get another service again.

Round 2

Reviewer 3 Report

appropriate for publication